# Relationship between Fatty Acids Composition/Antioxidant Potential of Breast Milk and Maternal Diet: Comparison with Infant Formulas

**DOI:** 10.3390/molecules25122910

**Published:** 2020-06-24

**Authors:** Michela Codini, Carmela Tringaniello, Lina Cossignani, Antonio Boccuto, Alessandra Mirarchi, Laura Cerquiglini, Stefania Troiani, Giuseppa Verducci, Federica Filomena Patria, Carmela Conte, Samuela Cataldi, Maria Rachele Ceccarini, Rita Paroni, Michele Dei Cas, Tommaso Beccari, Francesco Curcio, Elisabetta Albi

**Affiliations:** 1Department of Pharmaceutical Sciences, University of Perugia, 06126 Perugia, Italy; michela.codini@unipg.it (M.C.); carmela.tringaniello@unipg.it (C.T.); lina.cossignani@unipg.it (L.C.); alemirarchi@libero.it (A.M.); giuseppa.verducci@unipg.it (G.V.); patriafederica@gmail.com (F.F.P.); carmela.conte@unipg.it (C.C.); samuelacataldi@libero.it (S.C.); mariarachele.ceccarini@unipg.it (M.R.C.); tommaso.beccari@unipg.it (T.B.); 2Department of Mathematics and Computer Science, University of Perugia, 06126 Perugia, Italy; antonio.boccuto@unipg.it; 3Struttura Complessa di Neonatologia e Terapia Intensiva Neonatale– Azienda Ospedaliera Santa Maria della Misericordia—Perugia, 06129 Perugia, Italy; laura.cerquiglini@ospedale.perugia.it (L.C.); stefania.troiani@ospedale.perugia.it (S.T.); 4Department of Health Sciences, Università degli Studi di Milano, 20142 Milan, Italy; rita.paroni@unimi.it (R.P.); michele.deicas@unimi.it (M.D.C.); 5Department of Medicine (DAME), University of Udine, 33100 Udine, Italy; francesco.curcio@uniud.it

**Keywords:** human milk, fatty acids, PUFA, antioxidant potential, infant formulas

## Abstract

The fatty acid composition of human breast milk is relevant for the energy, immunity and eicosanoid production in infants. Additionally, the antioxidant properties of foods are essential for human health. Therefore, in the present study we aimed to investigate the relationship between maternal diet and fatty acids composition as well as the antioxidant potential of breast milk from donors to human milk bank of Perugia’s hospital, Italy. Results were compared with infant formulas. We observed increased levels of total fatty acids and, in particular, saturated and monounsaturated fatty acids in milk from mothers fed on a vegetable and fruit-rich diet compared with a Mediterranean diet. In the same milk, a reduced antioxidant potential was found. All infant formulas resulted in richer total fatty acid content than human breast milk. Only some formulas were qualitatively similar to breast milk. Of note, the antioxidant potential of the formulas was higher or lower than the human milk with the exception of one sample. The antioxidant potential of four formulas was very high. Dietary supplementation with antioxidants has been shown to have a teratogenic effect and to increase the formation of metastases in adult. There are no data on the effects of excess antioxidants in the infants, but the possibility that they can be harmful cannot be excluded.

## 1. Introduction

The lipids in human milk are organized as fat globules or droplets that are secreted by mammary epithelial cells. They include especially triglycerides, but also phospholipids, sphingolipids and cholesterol esters—all containing fatty acids (FAs). It is generally known to the scientific community that FAs can be distinguished as saturated (SFAs) or unsaturated fatty acids (USFAs). The latter can be monounsaturated fatty acids (MUFAs, one double bond in carbon chain) and polyunsaturated fatty acids (PUFAs, more than one double bond in carbon chain). Moreover, FAs are divided into four classes based on the length of the monocarboxylic chain: short-chain FAs (4 carbon atoms) useful as growth factors, medium-chain FAs (6–12 carbon atoms) available as energy source, long-chain FA (14–18 carbon atoms) useful as source of energy is they are SFAs and as molecules involved in fundamental metabolic processes of the cells if they are USFAs, and very long-chain FAs that act as functional molecules of biologic membranes with the role in cell fate (20–36 carbon atoms) [1]. In milk, short-chain FAs are responsible for its typical flavor; medium-chain FAs are important for its functional properties [2]. Human breast milk (HBM) contains—in addition to short and medium-chain FAs—long- and very long-chain FAs, especially arachidonic acid (ARA, 20:4n-6) and docosahexaenoic acid (DHA, 22:6n-3) [3]. The FAs of HBM provide energy and immunity, induce eicosanoid production and are used for body structure [4]. Thus, FAs are important for the physiology of development. 

It is of interest that a large variation of FA content in HBM is present across different populations, especially due to the total amount and quality of fat intake of pregnant and lactating mothers [5]. Nutritional errors during the last period of pregnancy and during breastfeeding induce changes in the composition of HBM, usually considered the optimum feeding regime for newborn infants [6]. In fact, HBM allows complete nutrition including bioactive health factors and its change may affect the physiological development of infants [7]. Moreover, HBM can be considered a living liquid able to mature and change over time, based on biologic and psychological needs of infants. Thus, milk produced in the first 72 h after childbirth or colostrum is transformed into transition milk and then into mature milk with the change in the content of water, proteins, lipids, vitamins and minerals. The macro- [8] and micronutrient [9,10] composition of milk is closely related to the maternal body composition and therefore to her feeding during pregnancy. The FA composition of HBM depends on the physiological status of the mother, maternal diet, endogenous biosynthesis in the mammary gland, and on the fat deposits of the mothers from which the FAs are released [11]. In milk, changes in FAs composition are usually used to highlight its oxidative stability [12] and, in particular, the level of USFAs [13]. In fact, their loss is connected with the peroxide value [14]. In accelerated oxidation, FAs are broken down to oxidation products which lead to the development of rancid flavor with the change for milk quality [15]. Notably, milk contains antioxidants distinguished in two categories, fat soluble and water-soluble antioxidants [16]. The first, include vitamin A, E and carotenoids and the second casein, vitamin C, cysteine, valine, lactase, glutathione peroxidase and superoxide dismutase, zinc and selenium [17]. It is relevant that milk antioxidant properties are species-specific [18]. Therefore, breastfeeding provides optimal antioxidant food for neonates [19].

Few studies have been conducted on the antioxidant capacity of HBM and on its comparison with infant formulas (IFs), used as substitutes of HBM when it is not available [20,21,22,23,24]. All findings above reported indicate the essential role of FAs and of antioxidant properties of HBM, but there are no studies that highlight, at the same time, the variations of the different parameters. Therefore, in the present study, we aimed to investigate: (1) the influence of diet on fatty-acid composition and antioxidant potential of breast milk; (2) the comparison between fatty-acid composition and antioxidant potential of infant formulas and those of breast milk. In this regard, milks formulated for preterm and term infants compared to milks of mothers with preterm and term birth were analyzed. These data are relevant because the composition of milk is different in relation to the gestational age [11] and because preterm infants have different nutritional needs in comparison with term infants.

## 2. Results

### 2.1. Mothers and Infants

It was reported that HBM composition depends on endogenous biosynthesis of molecules in the mammary gland, mother’s body mass index (BMI), physiological and nutritional status, gestational age at birth, diet and breed [5,9,10,11]. In order to further understand the relationships among different factors, we performed a study on milk from donor mothers in Umbria, Italy. For the study, we first analyzed parameters of mothers and infants with an interview. The exclusion criteria were: 1) mothers with a history of pathology; b) mothers who use drug, alcohol and smoke. The inclusion criteria were: 1) the BMI > 25 and BMI < 25 at the beginning of pregnancy; 3) physical activity carried out or not before and after pregnancy; 4) preterm birth (PT) and term birth (T). Thirty mothers aged 32.32 ± 4.89 were studied. The analysis of the interview indicated that of 30 considered mothers, 8 had a BMI > 25 at the beginning of pregnancy. The increase in weight during pregnancy was not related to the BMI (11.5 ± 3.4 kg for BMI > 25 and 12.50 ± 4.1 kg for BMI < 25). Only 4 mothers did physical activity during pregnancy (all belonged to a group with BMI < 25) and the increase in weight was moderately, but not highly significant related with it (10.5 ± 1.8 kg against 12.41 ± 3.8 kg for BMI < 25). Of note, the weight of infants was independent on the BMI of mothers (1.6 ± 3.1 kg for PT birth and 3.4 ± 4.1 kg for T birth in mothers with BMI > 25 and 1.52 ± 4.0 kg for PT birth and 3.3 ± 3.6 kg for T birth in mothers with BMI < 25). In addition, the diet did not influence in significant manner the weight of mothers during pregnancy and of infants at the birth. Different diets were considered: 1) Mediterranean diet (MD); diet rich in vegetables and fruit and with very low content of meat, fish, eggs and cereals (VF); diet rich in meat and fish with very low content of eggs, vegetables and fruit (MF); diet rich in meat and cereals with very low content of eggs, meat, vegetables and fruit (MC). Results showed that 18 mothers followed MD, 4 mothers followed VF diet, 4 mothers followed MF diet, and 4 mothers followed MC diet. The age of mothers was 31.66 ± 5.1 years for MD, 35.00 ± 1.26 years for VF, 33.2 ± 3.00 years for MF, 30.75 ± 3.27 years for MC diets. The increase in the weight of mothers during pregnancy was 13.33 ± 4.38 kg, 11.00 ± 2.34 kg, 11.6 ± 2.41 and 11.02 ± 3.74 with MD, VF, MF and MC diet, respectively. The age of mothers with PT was 32.12 ± 3.29 years and that of mothers with T was 31.71 ± 4.50. The weight of infants was 3.61 ± 0.39 kg, 3.41 ± 0.25 kg, 3.32 ± 0.64 kg and 3.09 ± 4.4 in T birth with MD, VF, MF and MC maternal diet, respectively. The PT infants from mothers fed on MD were 6 and weighted 1.77 ± 0.25 kg; none PT infant was from mothers fed on VF; 1 PT infant was from mothers fed on MD and weighted 1.95 kg; 1 PT infant was from mothers fed on MC and weighted 1.55 kg. Therefore, the type of diet is not relevant for the weight increase of mothers during pregnancy and for the weight of infants at the birth.

### 2.2. Fatty acid Composition of Human Breast Milk and Infant Formulas

First, we set out to measure FA composition of HBM of mothers fed on MD, VF, MF and MC diet in order to highlight, with accurate statistical analysis, the specific role of the diet. We considered the general similarity or difference of total FAs and SFAs–MUFAs–PUFAs in milk of mothers fed on VF, MF and MC diet in comparison with milk of mothers fed on MD) and then specific similarity or difference of each SFA–MUFA–PUFA species. Thus, MD was empirically used as comparison diet because it was used by most mothers. Intriguingly, the results showed that only VF diet impacted HBM FA composition. In fact, total FAs increased (Figure 1a) especially SFAs and MUFAs content (Figure 1b). Among SFA, short-chain, medium and long FAs increased significantly. Notably, palmitic acid (C18:0) increased in highly significant manner (Figure 1c). Among MUFAs, there were a specific increase of oleic acid (C18:1n9) and a reduction of other FAs (Figure 1d), so that the total MUFAs unchanged (Figure 1b).

Among PUFAs only DPH was highly reduced with VF. The reduction of total PUFAs with MC diet (Figure 1b) was due to the low level of n-6 PUFA (Figure 2a), specifically of linoleic acid (Figure 2c) with consequently lower level of n-6 PUFA/n-3 PUFA ratio (Figure 2d).

It then became important to determine whether FA composition of IFs was similar to the HBM of mothers fed on MD. Considering that on the market there are IFs for preterm (PT) and term (T) babies as these have different nutritional needs, to increase the power of comparisons, data of HBM from mothers fed on MD were grouped in PT and T. The rationale is that it is important to know the differences between IFs and HBM because it is good that IFs are as close as possible to HBM.

As can be seen, milk of T mothers had a lower level of total FAs (Figure 3a) in comparison with milk of PT mothers, due to the lower level of both SFAs and PUFAs (Figure 3b). All analyzed SFA species showed reduced values (Figure 3c). No important variations of MUFA were found (Figure 3c).

Among n-3 PUFA, you can see a reduction of alfa-linoleic acid and an increase of EPA (Figure 4b) that together did not change the total level of n-3 PUFA (Figure 4a). Differently, n-6 PUFA was moderately reduced because of the lower level of linoleic acid (Figure 4c).

Thus, we analyzed separately 5 IF used for PT neonates and 5 IF used for T neonates and compared the results with PT and T HBM, respectively. As can be seen, all PT formulas had a very higher content of total FAs than PT HBM, in particular IF 2 and 3 (Figure 5a). The IF 1 and 2 were very highly richer in SFAs and milk 3 in MUFAs; formula 5 was that most similar to HBM (Figure 5b). Formula 2 was rich in short, medium and long chain FAs (Figure 5c) and formula 4 in oleic acid (C18:1n9) (Figure 5d).

All formulas were rich in n-3 PUFA (Figure 6a) and, in particular, in alfa-linolenic acid and DHA content, with the exception of formula 1 (Figure 6b). Furthermore, n-6 PUFA had a high value, in particular in formulas 1 and 3 (Figure 6a) due to high level of linoleic and ARA content (Figure 6d). However, the high level of n-3 PUFA was responsible for the lower value of omega6/omega3 ratio in formulas respect to HBM.

The IF for T infants were all richer in total FAs than T HBM with the exception of formula 2 that is poorer (Figure 7a), due to the content of SFAs, MUFAs and PUFAs, (Figure 7b), each class rich in all its components (Figure 7c,d).

The same IFs had a higher total content of n-3 PUFA (Figure 8a) with higher level of alfa-linolenic and DHA and lower level of EPA (Figure 8b). Despite the increase of total and of each species of n-6 PUFA (Figure 8a,c), n-3 PUFA were prevalent so that the n-6 PUFA/n-3 PUFA ratio was lower than that present in T HBM (Figure 8d).

### 2.3. Antioxidant Potential Of Human Breast Milk and Infant Formulas

To further investigate the influence of diet on HBM as good food, we next analyzed its antioxidant potential. As shown in Figure 9a, VF diet was responsible for a reduction of antioxidant potential value in comparison with milk from mothers fed on MD and the contrary was obtained with MF and MC diet. PT and T birth did not influence the antioxidant potential of milk from mothers fed on MD (Figure 9b).

PT IF had all values different from respective HBM; in particular, formulas 2, 3 and 4 were 2.6, 2.1 and 1.7 times higher (Figure 10a). Among T IF, only formula 5 had a value similar to that of T HBM and formula 3 was 5 times higher (Figure 10b).

## 3. Discussion

Major revelations from this study are that: (1) VF diet is responsible for a high content of SFAs and MUFAs and a low antioxidant potential of HBM; (2) mothers fed on MD produced a milk rich in SFAs and PUFAs without changes of antioxidant potential if the infant was PT birth respect to T birth; (3) IFs are highly variable in the composition of FAs both from each other and from breast milk with truly significant variations in the antioxidant potential.

### 3.1. Relevance of Fatty Acids in Human Breast Milk

To date, research has established the importance of FA species for the nutritional characteristics of milk. FAs composition of HBM depends on many factors, as endogenous biosynthesis in the mammary gland, fat deposits of the mothers, gene predisposition and maternal diet [5,11]. We focused the attention on maternal diet because it is a modular factor. Moreover, it has been demonstrated that nutritional errors during the last period of pregnancy and during breastfeeding induce changes in the composition of HBM [6]. Since the foods introduced by mothers with the diet contain many macro- and micronutrients, it is really difficult to establish exactly which nutrients interfere with the lipid metabolism of the mammary gland. Our study shows that certainly the FA composition of HBM varied with VF, MF and MC diets towards MD. Yuan et al. [25] highlighted that lipid aggregates of HBM present at the end of gastric digestion become smaller and then resolve in two hours in intestinal digestion, while IFs present a more large number of aggregates that slow lipid digestion. Therefore, differences in lipid absorption are present between HBM and IFs. The fat aggregates are absent in vegetable oils [4]. Among FAs, SFAs are considered of lower value than PUFAs for human health. In fact, PUFAs are metabolically transformed in molecules involved in the regulation of different cell functions, the eicosanoids as prostaglandins, leukotrienes, thromboxane, resolvins, protectins, maresins [26]. The eicosanoids and endocannabinoids, also produced from PUFAs, are essential for the development and functioning of the nervous system [27]. Moreover, PUFAs have immunomodulatory properties, essential for the infants. In our study, the high level of linoleic and α-linolenic acid can be found with any type of diet, suggesting that it is a specific characteristic of HBM. Therefore, our data showing an enrichment of PUFAs in the IFs respect to HBM may indicate a positive aspect of IFs that may be very useful for the health of the neonates even if breastfeeding has been associated with optimal brain development and/or function [28]. Most IFs derive from cow milk, but they can be prepared also from vegetable oils. Begner et al. (2020) [29], in a pilot study, demonstrated that PT infants fed on exclusive HBM do not show in time severe cognitive developmental delay and present favorable growth and body composition. Differently, PT infants feed on IF show an increase in adverse outcomes respect to the use of HBM [30]. It is possible that it may be due to the presence of other nutrients specific for the brain in HBM.

### 3.2. Fatty Acids and Antioxidant Potential Relation in the Milk

Our data show that VF diet increases SFAs and MUFAs without changes of PUFAs, despite a reduced antioxidant potential compared to MD. On the other hand, milk from PT and T mothers have a similar antioxidant potential with different content of SFAs and PUFAs. It has been demonstrated that preservation of PUFAs, thanks to a good antioxidant potential, contributes to improve the quality of the milk [31]. It is possible that the antioxidant potential of HBM relatively influences the composition of FAs and in any case, despite being reduced, is capable of protecting PUFAs. Our data disagree with Oveisi et al. (2010) [21] who reported that an increased consumption of dairy products, fruits and vegetables, cereals and nuts induce an increase of the total antioxidant capacity of the HBM [21]. It is true that mothers belonging to our study did not eat cereals and nuts and that the use of cereals together to meat (MC diet) increases the antioxidant potential of the milk. In a systematic review on HBM from all non-vegetarian, vegetarian and vegan mothers, the nutritional values appear all comparable with the exception of FA composition and micro-components [32]. The authors observed that the dietary choices cannot be an exclusion criterion for donor candidates in human milk banks even if in some countries vegan lactating women as milk donors are excluded. The mothers object of this study were not vegetarian or vegan but followed a diet rich in vegetables and fruit with very low content of meat, fish, eggs and cereals (VF). Considering the low content of SFAs and the high content of vitamins in vegetables and fruit, we would have expected a result opposite to that obtained. It is possible that sugars present in these foods are metabolically used to synthesize SFAs useful for body energy, considering the low intake of other foods. Moreover, the antioxidant potential is low because of vegetable vitamins are lost with cooking, recommended procedure during pregnancy. The higher level of antioxidant potential in HBM of mothers fed on MF and MC diet suggest that it may be due to polypeptides or proteins, i.e., glutathione, catalase, etc. It is known that pasteurization, thermal processes applied for the preservation of milk by reducing/eliminating the bacteria, may impact on its lipid content and antioxidant characteristics [33]. In our study, pasteurization was performed either in HBM intended for the milk bank or in cow’s milk used for the preparation of IF. In addition, HBM samples were frozen for storage in the milk bank. Since the antioxidant potential of HBM is reduced by freezing [19], we supposed that it could occurs also in IFs. Thus, we diluted—in accordance with the instructions of the companies—the samples of IF powder. Then, we froze the milk to subject it to the same conditions as HBM. Therefore, our analyses were direct to know the characteristics of HBM and IF samples that had undergone the same treatments. Significant research has been devoted to highlight that PT infants born relatively deficient in antioxidant defenses and with increased oxidant stress [23]. Antioxidant capacity of HBM of mothers with PT birth is higher than that HBM of mothers T birth [34]. Our results show no variations of antioxidant potential in the milk of PT mothers in comparison with T mothers who followed a MD, supporting the important role of this kind of the diet for HBM composition. Moreover, it has been shown that HBM has better antioxidant capacity than IFs [20,21]. In time, IFs have been changed in relation to the needs of neonates and today their production is a field in constant evolution. Notably, antioxidant potential of IFs analyzed in the present study is higher or lower than that of HBM of mother fed on MD, with the exception of one sample that has a value very similar to that of HBM. The antioxidant potential of 4 formulas is really high, probably due to an enrichment with vitamin or others nutrients. It is known that consumption in adults of high level of antioxidants can be harmful to health [35]. In fact, it is negatively associated with metabolic syndrome [36], has teratogen effect [37] and can increase melanoma metastasis [38]. At the moment, there are no data on the effects of antioxidants in infants, but the possibility that they can be harmful to health cannot be excluded.

## 4. Materials and Methods

### 4.1. Materials

Anhydrous sodium sulfate, chloroform, hexane, methanol and potassium hydroxide were purchased from Carlo Erba Reagents (Milan, Italy). Supelco™ 37 component fatty acid methyl esters (FAME) mix, containing the methyl esters of 37 fatty acids was supplied by Supelco (Bellefonte, PA, USA).

### 4.2. Sample Collection Procedure

For this study, BLUD (Banca del Latte Umano Donato, Struttura Complessa di Neonatologia e Terapia Intensiva Neonatale—Azienda Ospedaliera Santa Maria della Misericordia—Perugia, Italy) provided human milk samples from thirty healthy donors. All mothers who had pathologies, such as hypo- or hyperthyroidism, diabetes, adrenal diseases, autoimmune diseases and alteration of the lipid profile in the blood were excluded. The project was approved by the Bioethics Committee of Perugia University (number 2018-05) and all procedures were performed accordingly. All donors signed informed consent. The milk samples were collected between March 1, 2019 and September 30, 2019 by using standardized procedures. Immediately after collecting, milk samples were submitted to Holder pasteurization that aims to rid milk of potentially harmful germs by heating it to 62.5 °C (145 °F) for half an hour, and then cooling it back down to 4–10 °C, a method used the world over to help ensure that the milk distributed by human milk banks is safe for infants to consume. Samples were then stored in a −20 °C freezer before analysis in order to know exactly the characteristics of the milk used to feed the infants.

### 4.3. Mothers Interview

A specific interview was carried out with the mothers to find out: (a) the state of health of the pregnant and lactating subject; (b) the consumption of drugs, alcohol, smoking during pregnancy and during breastfeeding; (c) the anthropometric measures pre- and post-pregnancy; (d) physical activity carried out before and after pregnancy; (e) gestational time and weight of the newborn; (f) nutritional habits of the subject during pregnancy e during breastfeeding. Regarding the diet followed, the interview aimed to have information on how many mothers followed a Mediterranean diet (MD), how many had eaten mainly vegetables and fruit (VF), how many had eaten mainly meat and fish (MF) and, finally, how many had eaten mainly meat and cereals (MC).

### 4.4. Lipid Extraction and Gas Chromatographic Analysis of Fatty Acids

Milk lipid fraction was extracted using a chloroform–methanol mixture (2:1, *v/v*), following the procedure previously reported [39]. The fatty acid methyl esters (FAME) of total lipids were prepared by transmethylation with methanolic KOH and analyzed by high-resolution gas chromatography. A DANI 1000DPC gas-chromatograph (Norwalk, CT, USA), equipped with a split–splitless injector and a flame ionization detector, was used. FAME separation was performed with a CP-Select CB for FAME fused silica capillary column (50 m × 0.25 mm i.d., 0.25 μm f.t.; Varian, Superchrom, Milan, Italy). The injector and detector temperatures were 250 °C. The oven temperature was 60 °C, held for 5 min then raised to 225 °C at 3 °C/min; the final temperature was held for 10 min. The chromatograms were acquired and processed using Clarity integration software (DataApex, Ltd., Prague, Czech Republic).

A standard solution containing 37 FAME was used to identify the individual fatty acids. The percentage of each FA was calculated using the peak area corrected with the respective correction factors [39]. HRGC analysis was carried out in triplicate.

### 4.5. Antioxidant Assay by Oxygen Radical Absorbance Capacity (ORAC)

The antioxidant capacity of umbrian LBB extract was determined using the ORAC method as previously reported [40]. We have chosen ORAC method because it is a robust and reliable method. In fact, the ORAC assay, other common measures of antioxidant capacity include ferric ion reducing antioxidant power and trolox equivalence antioxidant capacity assays and therefore is considered to be a preferable method because of its biologic relevance [41]. A duplicate extraction was performed for each sample and used to evaluate the lipophilic (L-ORACFL) and hydrophilic ORACFL (H-ORACFL) values. Evaluations of the lipophilic and hydrophilic ORACFL in the LBBs samples were performed separately, and the total antioxidant capacity (TAC) was calculated by adding the L-ORACFL and H-ORACFL values. The ORACFL assays were carried out on a FLUOstar OPTIMA microplate fluorescence reader (BMG LABTECH, Offenburg, Germany) at an excitation wavelength of 485 nm and an emission wavelength of 520 nm. The procedure was based on the method of Zulueta et al. (2009) with slight modifications. Briefly, 2,20-azobis (2-methylpropionamide) dihydrochloride (AAPH) was used as a peroxyl radical generator, trolox was used as a reference antioxidant standard, and fluorescein was used as a fluorescent probe. The data are expressed as micromoles of trolox equivalents (TE) per gram of sample (μmol TE/g).

### 4.6. Statistical Analysis

We first consider suitable samples of data concerning the concentration of different acids of milk samples, and successively the corresponding samples of data concerning the antioxidant potential. We used MD samples consisting of that data on a suitable number of patients, which better express the goodness of our experimental results. We compute the related mean and standard deviation by using Excel. Successively, we consider three other samples, VF, MF and MC, respectively, which contain the corresponding data about other smaller groups of patients. We compute again the mean and the standard deviation of each sample. Moreover, we called 1PT, 2PT, 3PT, 4PT and 5PT samples of preterm formulas from 5 different companies and 1T, 2T, 3T, 4T, and 5T samples of term formulas from other 5 different companies, we also get the mean and the standard deviation. We compare with the MD samples other HBM samples obtained by our experimental results. Moreover, MD samples were divided into preterm HBM (PT) and term HBM (T) samples and T samples were analyzed against PT. PT and T infant formulas were analyzed against MD PT and T HBM, respectively. Without loss of generality, we may assume that the total world population of nursing mothers has a “normal” distribution, and that the chosen samples are pairwise independent. Since the number of patients of each sample is less than 30 and, in general, the variances of the samples are not equal, we deal with Welch’s t-test [42]. When we compare two samples $S_{1}$ and $S_{2}$, this test defines the statistic t by means of the formula where $n_{i}$ is the size, $\bar{y_{i}}$ is the mean and $S_{i}$ is the standard deviation of $S_{i}$, i=1,2 and in general $S_{1}\neq S_{2}$.
(1)$$t=\frac{\bar{y_{1}}-\bar{y_{2}}}{\sqrt{\frac{S_{1}^{2}}{n_{1}}\;\pm \;\frac{S_{2}^{2}}{n_{2}}}}\;,$$

We will test the null hypothesis $\mu_{1}=\mu_{2}$ ($H_{0}$) versus the alternative hypothesis $\mu_{1}\neq \mu_{2}$ ($H_{1}$), where $\mu_{1}$ and $\mu_{2}$ are the means corresponding to the world nursing mothers populations in connection with the considered milk antioxidant powers or acid concentrations, related to $S_{1}$ and $S_{2}$, respectively. The studied test statistic is:(2)$$T=\frac{\bar{y_{1}}-\bar{y_{2}}-\left(\mu_{1}-\mu_{2}\right)}{\sqrt{\frac{S_{1}^{2}}{n_{1}}\;\pm \;\frac{S_{2}^{2}}{n_{2}}}}\;.$$

In considering artificial milks, when the mean and the variance of the corresponding $S_{2}$ are given, it is not necessary to know all elements of $S_{2}$, because $n_{2}=3$: indeed, by the definition of mean and sample standard deviation, $\bar{y_{2}}$ and $S_{2}$ are equal to the mean and the standard deviation, respectively, of the sample $T_{2}=\left\{\bar{y_{2}}-S_{2},\bar{y_{2}},\bar{y_{2}}\;\pm \;S_{2}\right\}$, Thus, in this case, without restriction, in our calculations we “replace” $S_{2}$ with $T_{2}$.

The Excel functions T.TEST and TEST.T compute directly the *p*-value related to T, that is the quantity
(3)$$2\cdot \min\;\{Pr(\{T\leq t|H_{0}\}),\;Pr(\{T\geq t|H_{0}\})\},$$
where the symbols in (3) denote the probability that the test statistic T is less (resp. greater) than or equal to t, when the null hypothesis $H_{0}$ is true since we deal with a two-tail hypothesis test. The p-value is the smallest significance level at which the obtained data lead to reject the null hypothesis. As we consider the significance levels $\alpha_{1}=0.05$, $\alpha_{2}=0.01$ or $\alpha_{3}=0.001,$ the null hypothesis will be rejected if and only if the p-value is less than or equal to $\alpha_{1}$ (or $\alpha_{2}\;\mathrm{or}\;\alpha_{3}$). Thus, a p-value $p\leq \alpha_{1}$ (or $p\leq \alpha_{2}\;\mathrm{or}\;p\leq \alpha_{3}$) indicates that the sample $S_{2}$ is significant (resp. highly significant, strongly significant) with respect to $S_{1}$ or equivalently that $S_{1}$ is significant (resp. highly significant, strongly significant) with respect to $S_{2}$.

When we compare our “white” sample $S_{1}$ with the corresponding sample $S_{2}$ related with an artificial milk, if we have the mean, but not the variance of $S_{2}$, we consider the statistic $\tilde{t}$ defined by
(4)$$\tilde{t}=\frac{\left(\bar{y_{1}}-\bar{y_{2}}\right)\;\sqrt{n_{1}}}{S_{1}}\;,$$
where $n_{1}$ and $S_{1}$ are the size and the standard deviation of $S_{1}$, respectively and $\bar{y_{i}}$ is the mean of $S_{i}$, i=1,2. The associated test statistic is
(5)$$\tilde{T}=\frac{\left(\bar{y_{1}}-\mu_{2}\right)\;\sqrt{n_{1}}}{S_{1}}\;,$$
where $\bar{y_{1}}$, $n_{1}$, $S_{1}$ and $\mu_{2}$ have the same meaning as above. We will test the null hypothesis $\mu_{1}=\mu_{2}$ versus the alternative hypothesis $\mu_{1}\neq \mu_{2}$. The variable $\tilde{T}$ is a “Student-type” t random variable with $n_{1}-1$ degrees of freedom. The Excel functions TDIST and DISTRIB.T compute the related p-value. Note that in this case, if we do the hypothesis that $S_{2}$ has null variance, then considering t (resp. T) is equivalent to dealing with $\tilde{t}$ (resp. $\tilde{T})$ and the involved statistic is independent of the size of $S_{2}$.

Now, let $S_{0}$ be the first “white sample” on the antioxidant power of the considered milks and let n, $\bar{X}$, S denote its size, mean and standard deviation, respectively. We compute a 95, a 99 and a 99.9 percent confidence interval estimator for the mean μ of the total world population of nursing mothers. Since the variance $\sigma^{2}$ of this population is unknown and the size of $S_{0}$ is less than 30, it is advisable to replace $\sigma^{2}$ with its estimator $S^{2}$, and to deal with the random variable
(6)$$T_{n-1}=\frac{\left(\bar{X}-\mu \right)\;\sqrt{n}}{S}\;,$$
which is a “Student-type” t random variable with n−1 degrees of freedom. Let us consider the probability density $f_{n-1}$ associated with $T_{n-1}$ and let $t_{n-1,\alpha_{i}}$ be the real number such that $Pr(\{T_{n-1}>t_{n-1,\alpha_{i}}\})=\alpha_{i}$, i=1,2,3. Since the graph of $f_{n-1}$ is symmetric with respect to the y-axis, we get
(7)$$\begin{array}{l} 1-\alpha_{i}=Pr\left(\left\{\left|T_{n-1}\right|\leq t_{n-1,\alpha_{i}/2}\right\}\right)=Pr\left(\left\{\frac{\left|\bar{X}-\mu \right|\;\sqrt{n}}{S}\leq t_{n-1,\alpha_{i}/2}\right\}\right)\\ =Pr\left(\left\{\bar{X}-t_{n-1,\alpha_{i}/2}\;\frac{S}{\sqrt{n}}\leq \mu \leq \bar{X}\;\pm \;t_{n-1,\alpha_{i}/2}\;\frac{S}{\sqrt{n}}\right\}\right). \end{array}$$

Thus, a $100\left(1-\alpha_{i}\right)$ percent confidence interval estimator for the mean μ of the considered population is given by
(8)$$\bar{X}\;\pm \;t_{n-1,\alpha_{i}/2}\;\frac{S}{\sqrt{n}}\;,$$
that is [43].
(9)$$\left[\bar{X}-t_{n-1,\alpha_{i}/2}\;\frac{S}{\sqrt{n}},\;\bar{X}\;\pm \;t_{n-1,\alpha_{i}/2}\;\frac{S}{\sqrt{n}}\right]$$

The computation of this interval is done by using Excel again. When we compare two samples $S_{1}$, $S_{2}$ and we get only the size, the mean and the standard deviation of the samples without having the complete data, like for instance in the case of the acids X, Y, Z, etc., it is not possible to use directly the Excel functions T.TEST or TEST.T. Therefore, in Welch’s test, we first compute the number of the degrees of freedom ν by means of the formula (formula 26) [41] and then we round ν to the nearest integer. The Excel functions TDIST and DISTRIB.T compute the corresponding p-value.
(10)$$\nu =\frac{{\left(\frac{S_{1}^{2}}{n_{1}}\;\pm \;\frac{S_{2}^{2}}{n_{2}}\right)}^{2}}{\frac{S_{1}^{4}}{n_{1}^{2}\left(n_{1}-1\right)}\;\pm \;\frac{S_{2}^{4}}{n_{2}^{2}\left(n_{2}-1\right)}}$$

## Figures and Tables

**Figure 1 molecules-25-02910-f001:**
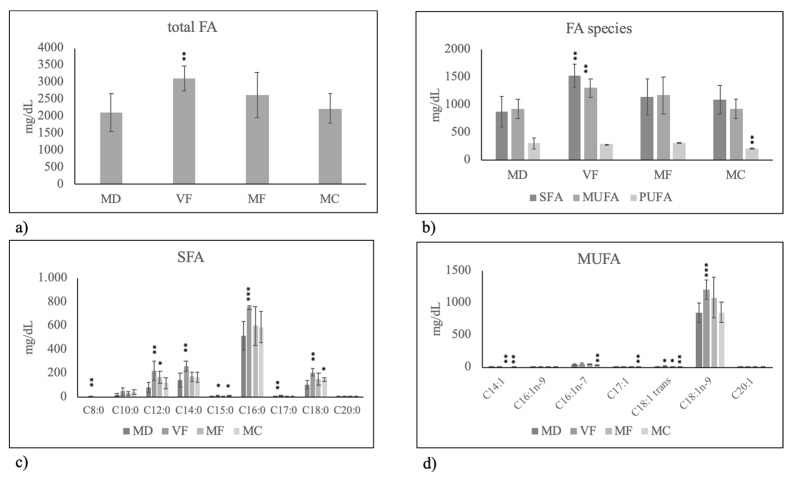
Fatty acid composition in human breast milk of mothers fed on different diets during pregnancy. MD, Mediterranean diet; VF, diet rich in vegetables and fruit and with very low content of meat, fish, eggs and cereals; MF, diet rich in meat and fish with very low content of eggs, vegetables and fruit; MC, diet rich in meat and cereals with very low content of eggs, meat, vegetables and fruit. (**a**) total fatty acids (FA); (**b**) Saturated fatty acids (SFA), monounsaturated fatty acids( MUFA) and polyunsaturated fatty acids (PUFA); (**c**) species of SFA; (**d**) species of MUFA. Data are expressed as mean ± SD calculated as reported in “Statistical analysis”. Significance of VF, MF, MC versus MD: * *p* < 0.05; ** *p* < 0.01; *** *p* < 0.001.

**Figure 2 molecules-25-02910-f002:**
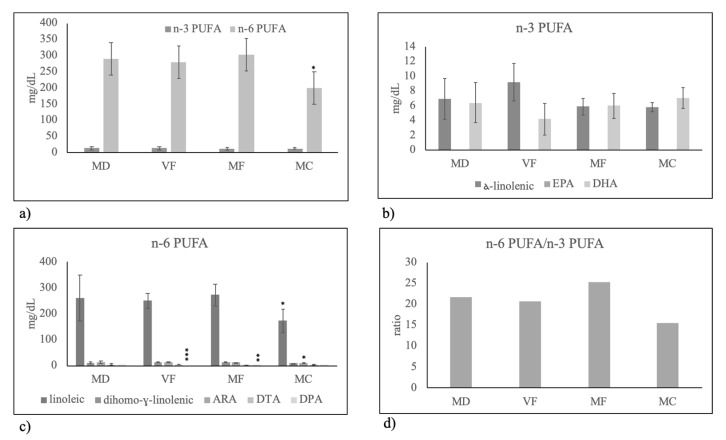
Polyunsaturated fatty acid composition in human breast milk of mothers fed on different diets during pregnancy. MD, Mediterranean diet; VF, diet rich in vegetables and fruit and with very low content of meat, fish, eggs and cereals; MF, diet rich in meat and fish with very low content of eggs, vegetables and fruit; MC, diet rich in meat and cereals with very low content of eggs, meat, vegetables and fruit. (**a**) total n-3 PUFA and n-6 PUFA; (**b**) species of n-3 PUFA: alfa-linolenic acid, eicosapentaenoic acid (EPA) and docosahexaenoic acid (DHA); (**c**) species of n-6 PUFA: linoleic acid, dihomo-gamma-linolenic acid, arachidonic acid (ARA), docosatetraenoic acid (DTA), docosapentaenoic acid (DPA); (**d**) n-6 PUFA/n-3 PUFA ratio. Data are expressed as mean ± SD calculated as reported in “Statistical analysis”. Significance of VF, MF, MC versus MD: * *p* < 0.05; ** *p* < 0.01; *** *p* < 0.001.

**Figure 3 molecules-25-02910-f003:**
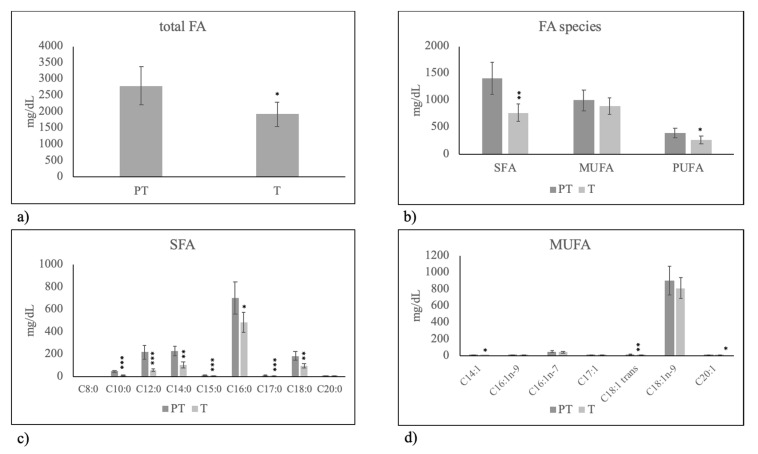
Fatty acid composition in human breast milk of mothers fed on Mediterranean diet with preterm birth (PT) and term birth (T). (**a**) total fatty acids (FA); (**b**) Saturated fatty acids (SFA), monounsaturated fatty acids (MUFA) and polyunsaturated fatty acids (PUFA); (**c**) species of SFA; (**d**) species of MUFA. Data are expressed as mean ± SD calculated as reported in “Statistical analysis”. Significance of T versus PT: * *p* < 0.05; ** *p* < 0.01; *** *p* < 0.001.

**Figure 4 molecules-25-02910-f004:**
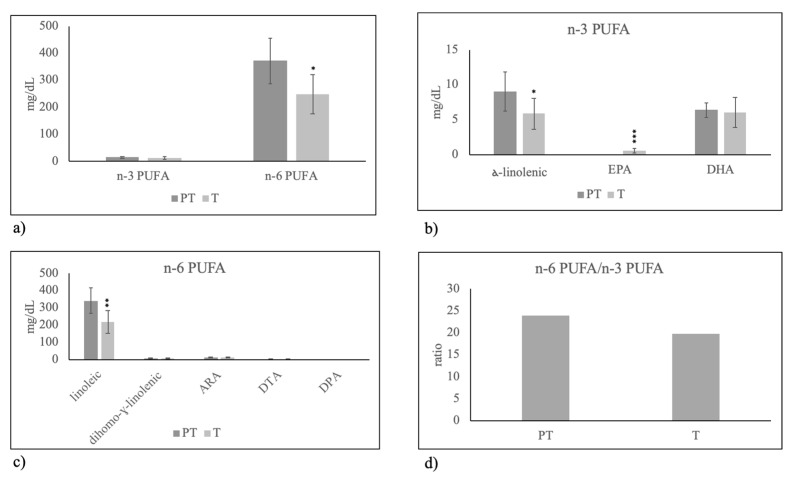
Polyunsaturated fatty acid composition in human breast milk of mothers fed on Mediterranean diet with preterm birth (PT) and term birth (T). (**a**) total n-3 PUFA and n-6 PUFA; (**b**) species of n-3 PUFA: alfa-linolenic acid, eicosapentaenoic acid (EPA) and docosahexaenoic acid (DHA); (**c**) species of n-6 PUFA: linoleic acid, dihomo-gamma-linolenic acid, arachidonic acid (ARA), docosatetraenoic acid (DTA), docosapentaenoic acid (DPA); (**d**) n-6 PUFA/n-3 PUFA ratio. Data are expressed as mean ± SD calculated as reported in “Statistical analysis”. Significance of T versus PT: * *p* < 0.05; ** *p* < 0.01; *** *p* < 0.001.

**Figure 5 molecules-25-02910-f005:**
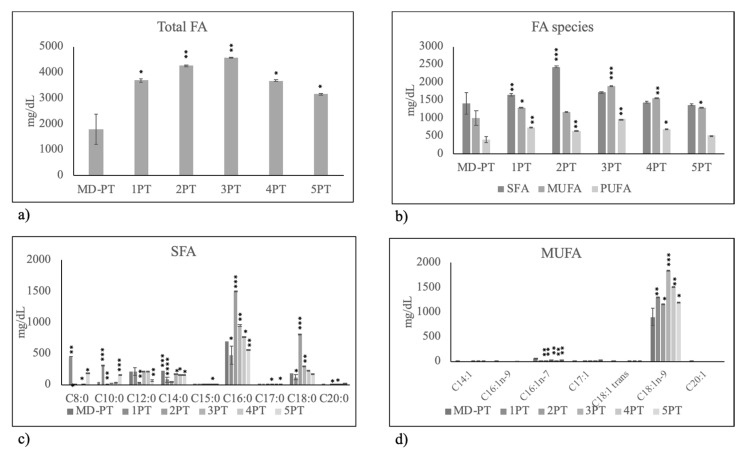
Fatty acid composition in human breast milk of mothers with preterm birth fed on Mediterranean diet (MD–PT) during pregnancy and in 5 infant formulas used to feed preterm birth infants. (**a**) total fatty acids (FA); (**b**) Saturated fatty acids (SFA), monounsaturated fatty acids (MUFA) and polyunsaturated fatty acids (PUFA); (**c**) species of SFA; (**d**) species of MUFA. Data are expressed as mean ± SD calculated as reported in “Statistical analysis”. Significance of 1PT, 2PT, 3PT, 4PT and 5PT versus MD–PT: * *p* < 0.05; ** *p* < 0.01; *** *p* < 0.001.

**Figure 6 molecules-25-02910-f006:**
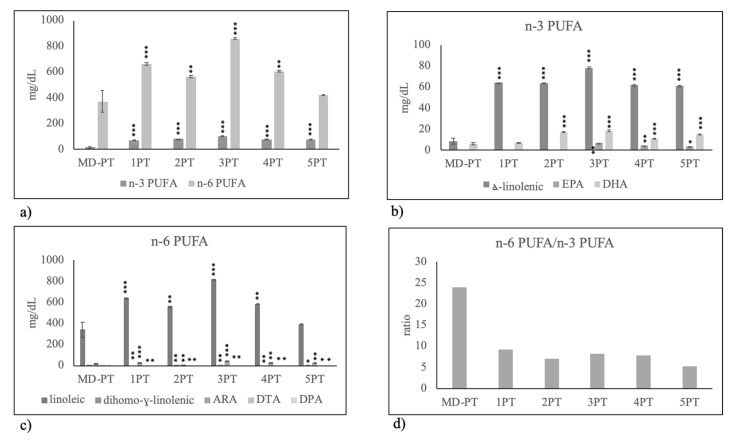
Polyunsaturated fatty acid composition in human breast milk of mothers with preterm birth fed on Mediterranean diet (MD–PT) during pregnancy and in infant formulas used to feed preterm birth infants (**a**) total n-3 PUFA and n-6 PUFA; (**b**) species of n-3 PUFA: alfa-linolenic acid, eicosapentaenoic acid (EPA) and docosahexaenoic acid (DHA); (**c**) species of n-6 PUFA: linoleic acid, dihomo-gamma-linolenic acid, arachidonic acid (ARA), docosatetraenoic acid (DTA), docosapentaenoic acid (DPA); (**d**) n-6 PUFA/n-3 PUFA ratio. Data are expressed as mean ± SD calculated as reported in “Statistical analysis”. Significance of 1PT, 2PT, 3PT, 4PT and 5PT versus MD–PT: * *p* < 0.05; ** *p* < 0.01; *** *p* < 0.001.

**Figure 7 molecules-25-02910-f007:**
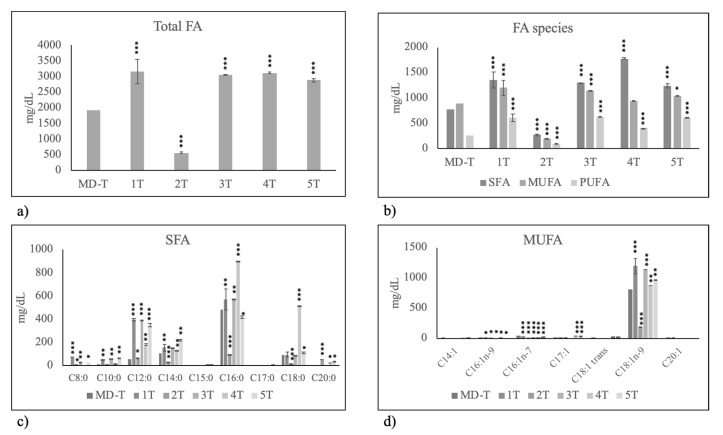
Fatty acid composition in human breast milk of mothers with term birth fed on Mediterranean diet (MD-T) during pregnancy and in 5 infant formulas used to feed term birth infants. (**a**) total fatty acids (FA); (**b**) Saturated fatty acids (SFA), monounsaturated fatty acids(MUFA) and polyunsaturated fatty acids (PUFA); (**c**) species of SFA; (**d**) species of MUFA. Data are expressed as mean ± SD calculated as reported in “Statistical analysis”. Significance of 1T, 2T, 3T, 4T and 5T versus MD-T: * *p* < 0.05; ** *p* < 0.01; *** *p* < 0.001.

**Figure 8 molecules-25-02910-f008:**
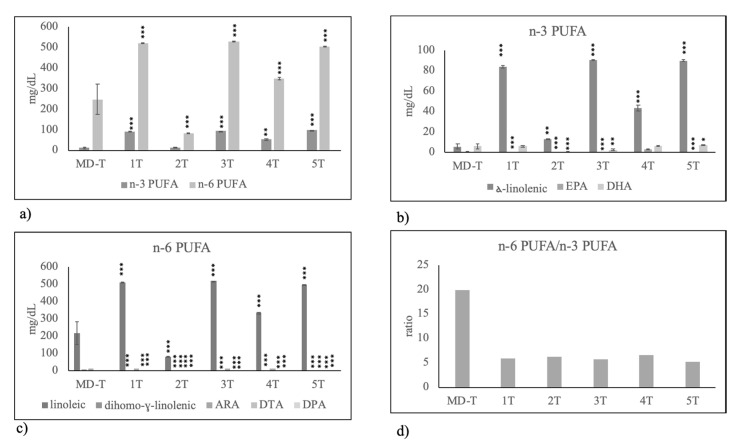
Polyunsaturated fatty acid composition in human breast milk of mothers with term birth fed on Mediterranean diet (MD-T) during pregnancy and in infant formulas used to feed term birth infants (**a**) total n-3 PUFA and n-6 PUFA; (**b**) species of n-3 PUFA: alfa-linolenic acid, eicosapentaenoic acid (EPA) and docosahexaenoic acid (DHA); (**c**) species of n-6 PUFA: linoleic acid, dihomo-gamma-linolenic acid, arachidonic acid (ARA), docosatetraenoic acid (DTA), docosapentaenoic acid (DPA); (**d**) n-6 PUFA/n-3 PUFA ratio. Data are expressed as mean ± SD calculated as reported in “Statistical analysis”. Significance of 1T, 2T, 3T, 4T and 5T versus MD-T: * *p* < 0.05; ** *p* < 0.01; *** *p* < 0.001.

**Figure 9 molecules-25-02910-f009:**
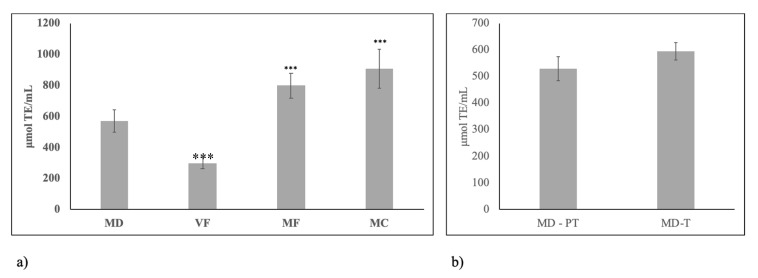
Antioxidant potential of human breast milk. (**a**) milk samples of mothers fed on different diets during pregnancy. MD, Mediterranean diet; VF, diet rich in vegetables and fruit and with very low content of meat, fish, eggs and cereals; MF, diet rich in meat and fish with very low content of eggs, vegetables and fruit; MC, diet rich in meat and cereals with very low content of eggs, meat, vegetables and fruit; (**b**) milk samples of mothers fed on Mediterranean diet during pregnancy with preterm birth (PB) and term birth (TB). Data are expressed as mean ± SD calculated as reported in “Statistical analysis”. Significance of VF, MF, MC versus MD: *** *p* < 0.001.

**Figure 10 molecules-25-02910-f010:**
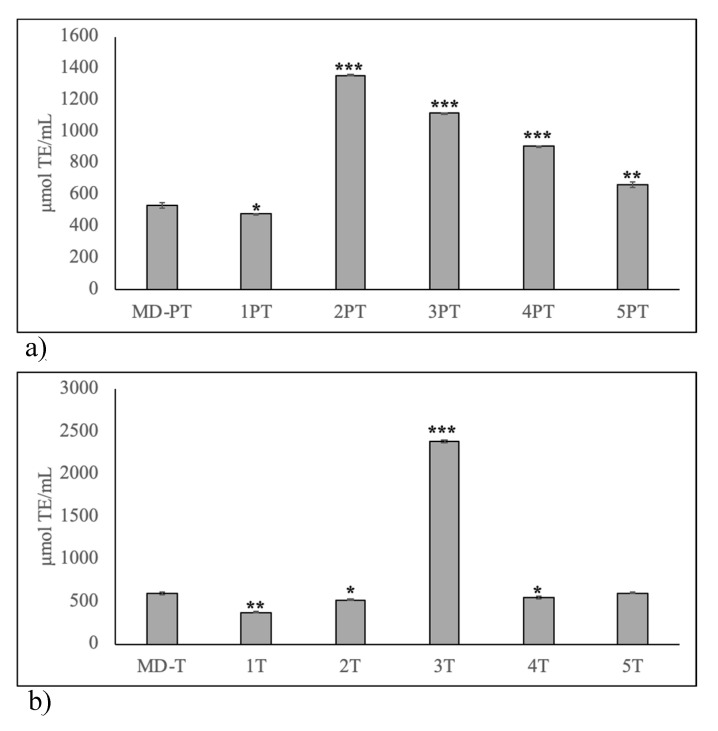
Comparison of antioxidant potential of human breast milk and infant formulas. (**a**) human breast milk of mothers with preterm birth (PT) fed on Mediterranean diet during pregnancy and in infant formulas used to feed preterm birth infants; (**b**) human breast milk of mothers with term birth (T) fed on Mediterranean diet during pregnancy and in infant formulas used to feed term birth infants. Data are expressed as mean ± SD calculated as reported in “Statistical analysis”. Significance of 1PT, 2PT, 3PT, 4PT and 5PT versus MD–PT and of 1 T, 2 T, 3 T, 4 T and 5 T versus MD-T: * *p* < 0.05; ** *p* < 0.01; *** *p* < 0.001.

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
