# Peer review of "Relationship between Fatty Acids Composition/Antioxidant Potential of Breast Milk and Maternal Diet: Comparison with Infant Formulas"

_molecules, 2020, doi:10.3390/molecules25122910_

Round 1
Reviewer 1 Report
Codini et al. investigated the composition of fatty acids in human breast milk after exposure to different diets, namely Mediterranean (MD), diet rich in fruits and vegetables with low meat, fish, egg and cereal intake (MF), diet rich in fruits and vegetables, meat, fish, egg and cereal (MC), and diet rich in meat and cereal with low egg, fruit and vegetable intake (MC). The group also compared the breast milk from mothers Pre-term and Full-term, and assessed the antioxidant capacity of the breast milk against infant formula.
This is an interesting study but needs improvement in the layout and explanation of the findings and discussion. It is difficult to follow the manuscript as there are several variables: 1) Diet; 2) Pre-term and Full-term; 3) Antioxidant capacity-this dilutes the objective of the study. For this reason, the objective of the study becomes ambiguous.
Please check English grammar and sentence constructions.
Comments:
- Please list out and abbreviation list. There are too many acronyms to follow. Either reduce it or make it very clear from the beginning with an abbreviation list.
- Line51: explain what the author means by building block?
- What is the reason for choosing Preterm subjects? Also, what is the rationale in the analysis to divide into 1T, 2T etc for both PT and T groups?
- Figures: In the statistic annotation, the authors used Significance *p<0.05, ….et What is this compared to?
- Omega 6 or omega 3 are not proper annotation for fatty acids. It should be omega-6 PUFA or n-6 PUFA etc.
- Figure 9: in the description what is the difference between MF and MC diet? Or MC is annotated incorrectly?
- Discussion, please divide into paragraphs.
- Line 291: Please explain, ‘Thus, we prepared and frozen also IFs’
- In my point of view, the findings of the antioxidant capacity is interesting with different diet. Unless I missed the point, the authors did not explain the connection of antioxidants and fatty acids in the breast milk.
- Can the authors explain/include the metabolism of the fatty acids from the diet and its relationship with breast milk?
- The authors find high linoleic acid and alpha-linolenic acid levels in the breast milk. What is the reason for this?
Author Response
This is an interesting study but needs improvement in the layout and explanation of the findings and discussion. It is difficult to follow the manuscript as there are several variables: 1) Diet; 2) Pre-term and Full-term; 3) Antioxidant capacity-this dilutes the objective of the study. For this reason, the objective of the study becomes ambiguous.
Thank you very much for this observation, The aim of the work has been revised at the end of “Introduction “ section (lines 112-120)
Please check English grammar and sentence constructions.
The English language has been revised
Comments:Please list out and abbreviation list. There are too many acronyms to follow. Either reduce it or make it very clear from the beginning with an abbreviation list.
This is a very good idea. A list of abbreviation has been included at the be beginning of the paper (lines 38-57)
Line51: explain what the author means by building block?
It has been revised (line 75)
What is the reason for choosing Preterm subjects? Also, what is the rationale in the analysis to divide into 1T, 2T etc for both PT and T groups?
The rationale for the analysis of preterm and term breast milk is that there are preterm and term formulated milks with different characteristics and it is good that these are as close as possible to breast milk. Therefore, the different types of infant formula used for preterm infants (PT1, PT2, etc.) have been compared with the milk produced by mothers with preterm birth fed on Mediterranean diet because their babies will be fed with this. So the different types of formulated milks used for term babies (T1, T2, etc.) were compared with the milk produced by mothers with term birth because their babies will be fed this.
It has been clarified (lines 194-198).
Figures: In the statistic annotation, the authors used Significance *p<0.05, ….et What is this compared to?
Thank you very much, it has been included in all figure legends
Omega 6 or omega 3 are not proper annotation for fatty acids. It should be omega-6 PUFA or n-6 PUFA etc.
Accordingly, omega 6 and omega have been changed along the text, in figures and in figure legends
Figure 9: in the description what is the difference between MF and MC diet? Or MC is annotated incorrectly?
The difference is that MF is a diet rich in meat and fish and MC is a diet rich in meat and cereals with very low content of …… meat……., as reported along the text
Discussion, please divide into paragraphs.
It has been made
Line 291: Please explain, ‘Thus, we prepared and frozen also IFs’
It has been made (lines 370,371)
In my point of view, the findings of the antioxidant capacity is interesting with different diet. Unless I missed the point, the authors did not explain the connection of antioxidants and fatty acids in the breast milk.
The antioxidant potential of human breast milk with the different diets is reported in the results, see fig.9a . The relationship between fatty acids and antioxidant power is reported in the discussion (line 341 and following)
Can the authors explain/include the metabolism of the fatty acids from the diet and its relationship with breast milk?
It has been explained in the discussion(lines 314-321)
The authors find high linoleic acid and alpha-linolenic acid levels in the breast milk. What is the reason for this?
It has been included in the discussion (lines 330-332)
Reviewer 2 Report
The work needs supplementation.
Title
“Relationship between fatty acids composition/antioxidant potential of breast milk and maternal diet: comparison with infant formulas”
Review:
Basic note: samples from 30 mothers were examined - on this basis, no statistical evaluation and conclusions can be made.
Abstract:
v. 29: “It is known that high consumption of antioxidants can be harmful in adults” - what does it mean? It is difficult to consume too many antioxidants in your diet. As you know, their bioavailability is quite low. Unless it concerns excessive consumption of dietary supplements?
Based on what research has this statement been made?
Material and Methods:
v.326: Why was the age of the mothers not taken into account?
Results:
87-114: It seems that the sample was too small to draw conclusions on this basis.
The work is very well developed. Research is very interesting and brings a lot to science. It seems, however, that they can only be treated as preliminary tests. On such a small group it is difficult to prove significant statistical results. The dependencies noted may in many cases be accidental. Therefore, drawing any conclusions from these studies is not justified. More methods should be used to assess antioxidant activity.
Author Response
Review:
Basic note: samples from 30 mothers were examined - on this basis, no statistical evaluation and conclusions can be made.
You are right that a number of 30 samples is not very high. However, the study for the collection of the samples lasted 4 months and required a lot of effort because all the mothers who had any type of clinical problem such as thyroid problems, diabetes, adrenal diseases, autoimmune diseases and alteration of the lipid profile in the blood were excluded. Healthy mothers had to give their consent to participate and gave their willingness to answer all the questions in a very complex questionnaire. Only a few of the healthy mothers made themselves available for the interview. For this reason, a statistician professor was included in the work. He adopted a statistical analysis suitable for the sample size. See “4.6. Statistical analysis” (lines 447-525). The exclusion criterion of mothers has been included in the text (lines 397-399).
Abstract:
29: “It is known that high consumption of antioxidants can be harmful in adults” - what does it mean? It is difficult to consume too many antioxidants in your diet. As you know, their bioavailability is quite low. Unless it concerns excessive consumption of dietary supplements?
Based on what research has this statement been made?
The statement has been corrected according to the studies reported in the “discussion” section (lines 32-35)
Material and Methods:
v.326: Why was the age of the mothers not taken into account?
thank you very much for this observation. Age was a parameter we hadn't considered. Instead this made us understand why not all women answered our questionnaire. These are mothers with a minimum of 27 years, with the exception of one mother of 19 and one of 24. So all mothers with a certain mental maturity. The other very young mothers did not participate. Thanks again. The ages were included in the manuscript (lines 130,131, 144,145, 147,148)
Results:
87-114: It seems that the sample was too small to draw conclusions on this basis.
The work is very well developed. Research is very interesting and brings a lot to science. It seems, however, that they can only be treated as preliminary tests. On such a small group it is difficult to prove significant statistical results. The dependencies noted may in many cases be accidental. Therefore, drawing any conclusions from these studies is not justified. More methods should be used to assess antioxidant activity.
We understand your observation and hope you can understand our explanation above in to “Review”. Thank you so much. Furthermore, our samples have all been used and we cannot do other analyzes on the same samples. Moreover, in this period it is not really possible to have contact with mothers and collect other milk samples. On the other hand, we have chosen ORAC method because it is a robust and reliable method. In fact, the ORAC assay, other common measures of antioxidant capacity include ferric ion reducing antioxidant power and Trolox equivalence antioxidant capacity assays and therefore is considered to be a preferable method because of its biological relevance (Jaffe, R.; Mani, J. Polyphenolics Evoke Healing Responses: Clinical Evidence and Role of Predictive Biomarkers in Polyphenols: Mechanisms of Action in Human Health and Disease Edited by: Ronald Ross Watson, Victor R. Preedy and Sherma Zibadi. ScienceDirect 2018, 29, 403-413).
It has been included in the text (lines 433-436).
Round 2
Reviewer 1 Report
Please check English writing. A few spotted with errors. Some words have no spacings (e.g. Lines 71, 81)
Line 20: Additionally
Line 26: In the same milk, reduced....
Line 163: n-3 PUFA (not omega 3)
Line 303: macro- and micronutrients
Line 381 hypo- or hyperthyroidism
Author Response
Please check English writing. A few spotted with errors. Some words have no spacings (e.g. Lines 71, 81)
Line 20: Additionally
Line 26: In the same milk, reduced....
Line 163: n-3 PUFA (not omega 3)
Line 303: macro- and micronutrients
Line 381 hypo- or hyperthyroidism
Thank you very much! all the work has been revised again
Reviewer 2 Report
Dear Authors,
I really find the work interesting, but 30 samples are not enough. Even the most outstanding statistics cannot draw conclusions based on such a small number of samples.
The ORAC method used to assess antiradical properties is actually quite stable. However, this method is quite specific and cannot be the only basis for drawing conclusions about the antiradical properties of the samples.
I understand that there are too few samples to carry out many analyzes. Maybe you need to change your research plan?
I believe that research should be supplemented.
Author Response
I really find the work interesting, but 30 samples are not enough. Even the most outstanding statistics cannot draw conclusions based on such a small number of samples.
I understand your position well, but it is really very difficult to expand the number of samples right now. On the other hand, numerous articles published this year have a number of milk samples similar to or less than ours. Below you can see only a few examples:
25 samples:
Longitudinal Variation of Amino Acid Levels in Human Milk and Their Associations with Infant Gender.
van Sadelhoff JHJ, van de Heijning BJM, Stahl B, Amodio S, Rings EHHM, Mearin ML, Garssen J, Hartog A.Nutrients. 2018 Sep 5;10(9):1233. doi: 10.3390/nu10091233
16 samples:
Free glutamine and glutamic acid increase in human milk through a three-month lactation period.
Agostoni C, Carratù B, Boniglia C, Lammardo AM, Riva E, Sanzini E.J Pediatr Gastroenterol Nutr. 2000 Nov;31(5):508-12. doi: 10.1097/00005176-200011000-00011
27 samples
Comparing Gut Microbiome in Mothers' Own Breast Milk- and Formula-Fed Moderate-Late Preterm Infants.
Wang Z, Neupane A, Vo R, White J, Wang X, Marzano SL.Front Microbiol. 2020 May 26;11:891. doi: 10.3389/fmicb.2020.00891. eCollection 2020
30 samples
Profiles of Human Milk Oligosaccharides and Their Relations to the Milk Microbiota of Breastfeeding Mothers in Dubai.
Ayoub Moubareck C, Lootah M, Tahlak M, Venema K.Nutrients. 2020 Jun 9;12(6):E1727. doi: 10.3390/nu12061727
33 samples
Milk-free diet followed by breastfeeding women.
Januszko P, Lange E.Rocz Panstw Zakl Hig. 2020;71(2):181-189. doi: 10.32394/rpzh.2020.0118.
10 samples
Untargeted lipidomics using liquid chromatography-ion mobility-mass spectrometry reveals novel triacylglycerides in human milk.
George AD, Gay MCL, Wlodek ME, Trengove RD, Murray K, Geddes DT.Sci Rep. 2020 Jun 9;10(1):9255. doi: 10.1038/s41598-020-66235-y.
20 samples
Human milk immunomodulatory proteins are related to development of infant body composition during the first year of lactation.
Gridneva Z, Lai CT, Rea A, Tie WJ, Ward LC, Murray K, Hartmann PE, Geddes DT.Pediatr Res. 2020 May 21. doi: 10.1038/s41390-020-0961-z. Online ahead of print
18 samples
Total metal content and chemical speciation analysis of iron, copper, zinc and iodine in human breast milk using high-performance liquid chromatography separation and inductively coupled plasma mass spectrometry detection.
Trinta VO, Padilha PC, Petronilho S, Santelli RE, Braz BF, Freire AS, Saunders C, Rocha HFD, Sanz-Medel A, Fernández-Sánchez ML.Food Chem. 2020 Oct 1;326:126978. doi: 10.1016/j.foodchem.2020.126978. Epub 2020 May 6.
20 samples
Maternal Allergy and the Presence of Nonhuman Proteinaceous Molecules in Human Milk.
Dekker PM, Boeren S, Wijga AH, Koppelman GH, Vervoort JJM, Hettinga KA.Nutrients. 2020 Apr 22;12(4):1169. doi: 10.3390/nu12041169.
31 samples
The Effect of Physical Activity on Human Milk Macronutrient Content and Its Volume.
Be'er M, Mandel D, Yelak A, Gal DL, Mangel L, Lubetzky R.Breastfeed Med. 2020 Jun;15(6):357-361. doi: 10.1089/bfm.2019.0292. Epub 2020 Apr 8
12 samples
Characterization of Extracellular Vesicles Isolated From Human Milk Using a Precipitation-Based Method.
Bickmore DC, Miklavcic JJ.Front Nutr. 2020 Mar 13;7:22. doi: 10.3389/fnut.2020.00022. eCollection 2020
The ORAC method used to assess antiradical properties is actually quite stable. However, this method is quite specific and cannot be the only basis for drawing conclusions about the antiradical properties of the samples.
I understand that there are too few samples to carry out many analyzes. Maybe you need to change your research plan?
I believe that research should be supplemented.
Of course, you are right! however we do not want to evaluate all the antiradical properties but only the antioxidant potential. In addition, the paper was considered with "minor revision" to be submitted within 5 days. Your comments will surely be very useful for our future work and for this we thank you very much